# New Generation of Fixture–Abutment Connection Combining Soft Tissue Design and Vertical Screw-Retained Restoration: 1-Year Clinical, Aesthetics and Radiographic Preliminary Evaluation

**DOI:** 10.3390/dj9040035

**Published:** 2021-03-24

**Authors:** Francesco Mattia Ceruso, Irene Ieria, Mirko Martelli, Aurea I. Lumbau, Erta Xhanari, Marco Gargari

**Affiliations:** 1Department of Dentistry “Fra G.B. Orsenigo—Ospedale San Pietro F.B.F.”, 00100 Rome, Italy; irene.ieria@gmail.com (I.I.); mirk.marte@libero.it (M.M.); marco.gargari@gmail.com (M.G.); 2Department of Periodontology and Implantology, University of Sassari, 07021 Sassari, Italy; alumbau@uniss.it; 3Implantology and Prosthetic Aspects, Master of Science in Dentistry Program, Aldent Univesity, 1022 Tirana, Albania; ertaxhanari@hotmail.com; 4Department of Clinical Science and Translational Medicine, University of Rome “Tor Vergata”, 00100 Rome, Italy

**Keywords:** dental implants, aesthetics, BOPT, Prama, marginal bone loss

## Abstract

Implant design factors and the abutment connection are correlated with crestal bone stability. The aim of the present study was to evaluate a new type of screw-retained prostheses delivered on tissue-level implants with conical external vertical seal and internal hexagon connection. Implants 4.25 mm in diameter and 10 mm in length (Prama, Sweden and Martina) were placed in partially edentulous patients needing at least one implant in the healed site, having sufficient bone volume. The implant neck was positioned above the bone crest. A healing abutment was placed according to a one-stage approach. Outcome measures were implant and prosthesis survival rate, any complications, marginal bone loss (MBL), periodontal parameters, and pink esthetic score (PES). Overall, 13 patients (4 women and 9 men; mean age 50 ± 22 years) with the same number of implants were treated and followed for one year after loading. At the 12-month follow up, no implant and no prosthesis failed, and no complications were experienced. The mean MBL experienced at the one year follow-up was 0.65 ± 0.48 mm. One year after loading, 2 out of 13 implants present bleeding on probing (15.4%), 4 out of 13 patients presented with plaque at the one year of follow-up (30.8%) and the PES was 10.5 ± 2.3 mm. Within the limitations of the present study, the analyzed implants seem to be a viable treatment option for the rehabilitation of a single tooth gap.

## 1. Introduction

Schroeder and co-authors showed through photomicrographic images the direct contact between the metal implant surface and the bone [1]. Since the mid-1970s, most of the scientific research has focused on the various types of implant surfaces that could increase the percentage of bone-to-implant contact (BIC) [2]. In fact, this research has allowed for modifying the BIC, bringing it to ever higher percentages, thus allowing faster osseointegration and greater implant stability quotients. Today, considering the important advances in the area of implant surfaces and consequently higher predictability of implant-prosthetic therapy, scientific research has shifted to maintaining the health of peri-implant tissues and to improving aesthetics associated with implant therapy. Incorrect management of the peri-implant soft tissues could affect the medium- and long-term clinical and radiographic outcomes. In fact, a bacterial colonization of the peri-implant tissues could evolves from simple mucositis into a complex peri-implant disease when peri-implant bone is resorbed, finally compromising functional and aesthetic success [3,4].

The possible causes of crestal bone resorption have been widely discussed in the literature, and many factors that control crestal bone levels around dental fixture have been examined [5,6,7,8,9,10]. To maintain the crestal bone stability, clinicians must understand that prosthetic as well as surgical aspects are responsible for the success of the treatments [10]. In fact, the biomechanical aspects refer to masticatory load as well as pathological jaw movements (i.e., bruxism) as a major cause, while the biological hypothesis considers the bacteria present at the level of the microgap between fixtures and abutments as main etiological agents. Hence, there is a need to introduce new prosthetic concepts able to improve esthetics and to reduce peri-implant bone loss. 

The new generation of fixture–abutment connection featured into the Prama implant (Sweden and Martina, Due Carrare, PD, Italy) is characterized by a transmucosal neck with a cylindrical path of 0.8 mm and a hyperbolic geometry portion of two mm designed to simulate an element prepared with the Biologically Oriented Preparation Technique (B.O.P.T.) [11,12,13]. These characteristics, associated with a screwed prosthesis, could promote the health of peri-implant soft and hard tissues and could reduce the marginal bone loss.

The purpose of the current case series study was to examine a new type of screw-retained prosthesis delivered on tissue-level implants with conical external vertical seal and internal hexagon connection [14,15,16,17]. The study was written according to the STROBE (Strengthening the Reporting of Observational Studies in Epidemiology) guidelines.

## 2. Results

A total of 15 patients were originally enrolled and consecutively treated, but two patients were lost at the one-year follow-up for reasons not related to the study. When called on the phone, both patients refused to undergo the one-year follow-up visit due to the COVID-19 pandemic. Overall, 13 patients were followed for one year after loading. The age of the patients was 50 ± 22 years. Of these, four patients were women and nine were men, and four patients smoked up to 10 cigarettes per day. During the entire follow-up period, no implants and no prostheses failed, and no complications were experienced, scoring a cumulative implant and prosthesis survival and success rate of 100%. The mean marginal bone loss (MBL) experienced during the first year after function was 0.65 ± 0.48 mm (95% CI from 0.40 to 0.91 mm).

One year after loading, 2 out of 13 implants presented bleeding on probing (0.15 ± 0.38; corresponding to the 15.4% of the sites). These and two others (a total of four out of 13 patients) presented with plaque at the one year of follow-up (0.31 ± 0.48; corresponding to the 30.8% of the sites). A possible reason why almost one third of the patients have unsatisfactory hygiene could be that patients received delayed professional maintenance therapy due to the COVID-19 pandemic.

At the one-year examination, the pink esthetic score was 10.46 ± 2.30 mm (95% CI from 9.26 to 11.66 mm).

## 3. Discussion

A conical external connection has been introduced for tissue-level implants. Due to the prospective case series study design, the main limitations were the lack of a control group and the small sample size. In addition, the gingiva biotype was not considered in the present study. Hence, this research should be considered as a proof-of-concept study to act as a pilot for future multicenter randomized controlled trial with a larger sample and a longer follow-up.

The results of the current study were partially in agreement with a similar publication by Canullo et al. [16] In both studies, all implants were clinically osseointegrated, with no sign of infection. The mean marginal bone loss experienced in the presented research was slightly higher compared to Canullo et al. [16]; however, the bone resorption was measured using the smooth collar as a reference. On the contrary, in the presented research, the transition between the golden transgingival neck and the treated implant surface was considered as a reference point. 

One of the prerequisites for a successful implant-supported restoration is the long-term maintenance of hard and soft tissue stability. For the latter, several factors influence the peri-implant tissue stability, summarized in local and systemic factors, as well as surgical and restorative techniques and materials [18,19,20]. Within these, the microgap between the implant and the abutment is a critical factor in the marginal bone loss. This represents on the one hand a source of bacteria contamination, resulting in a chronic inflammatory state, and on the other hand a source of micro-movement [21]. For these reasons, when the implant-abutment connection is located at the crestal bone level, it may result in marginal bone loss.

Several implant designs have been proposed to reduce the bacterial infiltration at the implant–abutment interface. In order to reduce this phenomenon, referring to the bone level implants, the change of fixture–abutment connections and the platform switching concept has been suggested [22]. The rationale of this system is to relocate the junction away from the bone in a horizontal direction.

The tissue-level implant was introduced with the aim to relocate the interface, with the associate bacterial infiltration, over the crestal bone [22,23]. Otherwise, the typical convergent design of the transmucosal collar leads to difficulty in the management of soft tissues, in particular in the anterior sites, where the aesthetic demands are higher. In addition, when placed in a more apical position, the convergent design can lead to a biological problem of bone reabsorption, due to compression of the bone [16,24].

Recently, an alternative soft tissue implant has been introduced to the market. The Prama system has been developed to transfer the so-called Biologically Oriented Preparation Technique from the prosthesis on natural teeth to the prosthesis on implants [13,18]. This presents a unique shape design of the collar that is connected to the corresponding tapered abutment, resulting in a feather-edge design with no finishing line, guaranteeing continuity with the post. In particular, the transmucosal neck is distinguished by a cylindrical path (0.8 mm of high) and a hyperbolic geometry portion (2 mm of high) [11,12,13]. This novel transmucosal design has been shown to provide advantages in maintaining the remodeling and stability of the soft and hard tissue [12,13,14,15,16]. In fact, the characteristic convergent shape of the collar reduces the pressure on the peri-implant tissue, transferring the Biologically Oriented Preparation Technique (BOPT)to the implant mucosa, which is adapted to the restoration.

Finally, in the present study, in addition to the newly developed, convergent collar design, the use of a screw-retain restoration allowed us to reduce the risk of MBL that could be generated by the incomplete removal of cement, contributing to a reduced risk of biological complications.

## 4. Material and Methods

The present research was designed as a prospective case series study aimed to evaluate a new generation of fixture–abutment connection combining soft tissue design and vertical screw-retained restoration. From March 2018 to May 2019, any partially edentulous patient in need of at least one implant in a healed site, 18 years or older and able to sign informed consent, was considered eligible for this study. This research adhered to the principles in the Declaration of Helsinki of 2008. Medical data were anonymized so that patients could not be identified. All subjects were informed about the study protocol and signed informed consent. Patients were treated at the Department of Dentistry "Fra G.B. Orsenigo-Ospedale San Pietro F.B.F.", University of Rome “Tor Vergata”, Rome, Italy. All the surgical and prosthetic procedures were performed by an expert clinician (F.M.C.). The present publication was approved by the ethics committee of Aldent University in Tirana (Protocol n°1/2021). Patients were not admitted to the study if any of the following exclusion criteria were present:General medical contraindications to implant surgery;Patients irradiated in the head and neck area in the last 5 years;Immunosuppressed or immunocompromised patients treated or under treatment with intravenous amino-bisphosphonates;Untreated periodontitis;Poor oral hygiene and motivation (bleeding and probing and/or plaque index ≤25%);Uncontrolled diabetes;Pregnancy or nursing;Substance abuser;Heavy smokers (more than 11 cigarettes/day);Psychiatric contraindications or unrealistic expectations;Patients with inflammation in the area intended for implant placement.

A preoperative cone-beam computer tomography (CBCT) scan was obtained for every potentially eligible patient to quantify bone volumes at the planned implant sites. A week before implant placement, all patients were subjected to professional oral hygiene. On the day of the surgery, all the patients rinsed with 0.2% chlorhexidine mouthwash for one minute. Two grams of amoxicillin were administered one hour before implant placement. Patients allergic to penicillin were given Clyndamicin 600 mg one hour before implant placement. Local anesthesia was induced using Articain with adrenaline 1:100,000. After crestal incision and full-thickness flap elevation, the implant site was prepared. Drills with increasing diameters were used to prepare the implant sites following the standard procedures as recommended by the manufacturer (Sweden and Martina). Bone quality was intraoperatively assessed and reported as hard, medium and soft. A drilling portal was adapted to the bone quality. The surgical unit was set with a torque of 35 Ncm during implant insertion. Back and forth movements were performed to place implants without exceeding 35 Ncm.

Implants of 10 mm in length and 4.25 mm in diameter (Prama, Sweden and Martina) were positioned with the implant neck above the bone crest. The implants used were characterized by a straight cylindrical section of 0.80 mm of high, followed by a section with truncated hyperbolic cone shape (truncated cone) two mm high, specifically designed to guarantee continuity with the post (Figure 1) by improving the contact area.

The implant neck also underwent an anodic passivation process that gave it the characteristic golden pale yellow color. The so-called ZirTi body of the implant was sand-blasted with zirconium and acid-etched with mineral acids (Figure 2).

A healing abutment was placed according to a one-stage approach, and the flap was closed with a resorbable 4.0 suture. Periapical radiographs of the study implants were taken. Patients were instructed to have a soft and cool diet for 1 week and to rinse with 0.2% of chlorhexidine mouthwash for one minute twice a day for 14 days and were not allowed to wear any removable denture, which could load the study implants. Sutures were removed after 7 to 10 days, and oral hygiene instructions were delivered. Ibuprofen 400 mg was prescribed to be taken if needed.

Four months after implant placement (Figure 3A,B), a customized open-tray impression with screw-retained transfer impression copings was taken at implant level using a polyether material (ImpregumTM, 3M ESPE, Seefeld, Germany). The screw-retained prosthetic crowns were delivered within a month. The occlusal surface was kept in slight contact with the opposite dentition. Periapical radiographs and clinical pictures of the study implants were taken, oral hygiene instructions were delivered, and patients were enrolled in a follow-up program. (Figure 3C–E).

Professional maintenance was delivered every 6 months after initial loading. Dental occlusion was evaluated at each follow-up visit. Patients were followed for at least one year after prosthesis delivery.
The main outcome measures were the success rates of the implants and prostheses, and any surgical and prosthetic complications that occurred during the entire follow-up were taken into account.An implant not osseointegrated, tested by tapping or rocking the implant head with a hand instrument, and any signs of radiolucency and/or fracture on an intraoral radiograph were considered a failure.A prosthesis was assumed to be unsuccessful when it needed to be replaced by an alternative prosthesis.Any biological (pain, swelling, suppuration, etc.) and/or mechanical complications (fracture of the framework and/or the veneering material, screw loosening, etc.) were examined.

Secondary outcomes were: marginal bone loss (MBL), periodontal parameters such us bleeding index (BI) and plaque index (PI), and pink esthetic score (PES).
MBL variation was evaluated using intraoral digital periapical radiographs at the implant loading (5 months after implant placement) and at the end of the first year on function (one year after loading). Intraoral radiographs were taken with the parallel technique by means of periapical radiographs with customized holders and were accepted or rejected for evaluation based on the clarity of the implant threads. All the readable ones were displayed in an image analysis program and evaluated under standardized conditions. The software has been calibrated for every single image using the known distance of the implant diameter or length. The distance from the reference point at the implant neck (transition between the golden transgingival neck and the treated implant surface, Figure 4) to the first bone to implant contact was taken as the horizontal marginal bone level at both mesial and distal aspects. The average radiographic values of mesial and distal measurements were taken for each implant. The difference between the marginal bone levels at various timepoints was taken as MBL. An independent radiologist not involved in the study performed all the bone measurements.BI and PI around the implant/abutment interfaces were estimated yearly using a plastic periodontal probe (Plast-o-Probe, Dentsply Maillefer, Ballaigues, Switzerland). The BI was evaluated around each implant as the presence of bleeding elicited 20 s after the careful insertion of the periodontal probe one mm into the mucosal sulcus, parallel to where the abutment (0 = no bleeding; 1 = bleeding). The PI, defined as the presence of plaque (0 = no plaque; 1 = plaque) on the abutment/restoration complex, measured by running the periodontal probe parallel to the abutment surfaces, and scored at one site for the implant. All periodontal measurements were carried out by an independent dental hygienist.The PES as aesthetic evaluation [25] was assessed by collecting pictures on vestibular and occlusal views, including at least 1 adjacent tooth per side. The values were determined at one year after the definitive loading visit. Seven variabilities (mesial papilla, distal papilla, soft tissue level, soft tissue contour, alveolar process deficiency, soft tissue color and texture) were assessed on a 0 to 2 score (0 being poorest and 2 being the best) by an independent outcome assessor not previously involved in the study (I.I.).

## 5. Statistical Analysis

No sample size was calculated. All data analysis was carried out according to a pre-established analysis plan by a biostatistician with expertise in dentistry. Data were collected using mean and standard deviation, including confidence interval. Comparisons between time points and the baseline measurements were made by paired t-tests, to detect any changes in marginal peri-implant bone levels. All statistical comparisons were conducted at the 0.05 level of significance.

## 6. Conclusions

With the limitations of the present study, the newly developed implants with a fixture-abutment connection combining soft tissue design and vertical screw-retained restoration seem to be a viable treatment option for the rehabilitation of a single tooth gap. The positive outcome should encourage further study to confirm these preliminary results. Further studies are needed to confirm these preliminary results.

## Figures and Tables

**Figure 1 dentistry-09-00035-f001:**
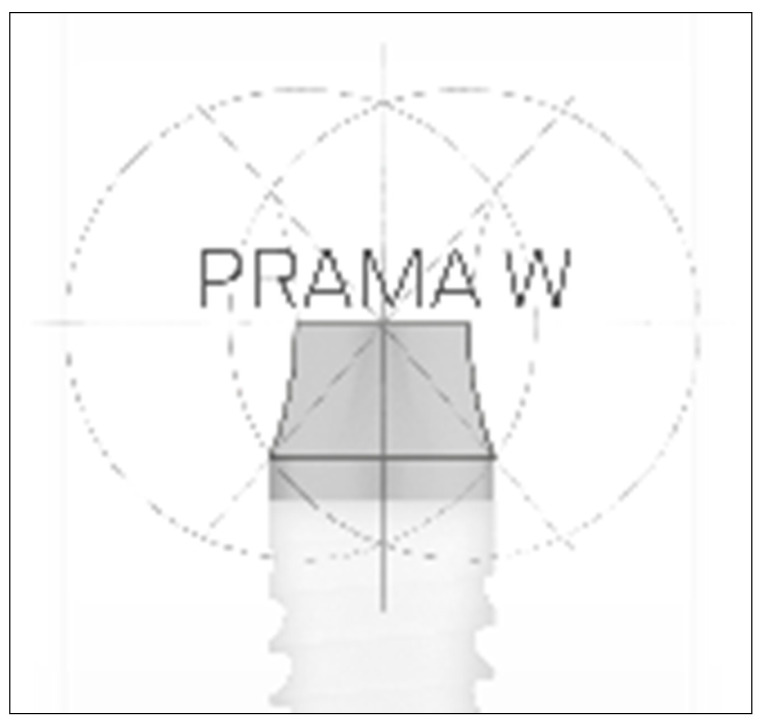
Geometry of the implant neck.

**Figure 2 dentistry-09-00035-f002:**
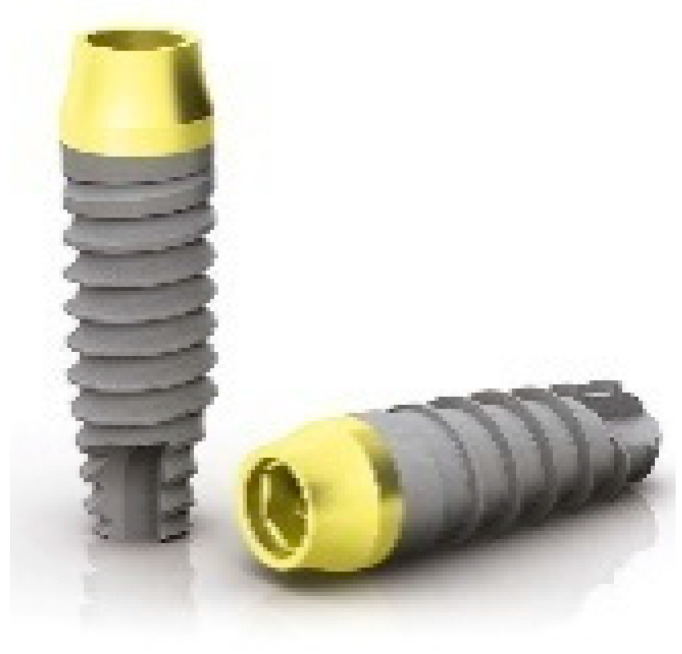
Prima implants (Sweden and Martina).

**Figure 3 dentistry-09-00035-f003:**
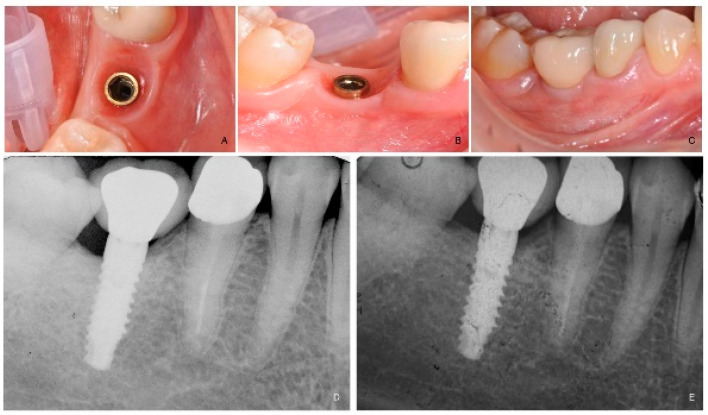
(**A**): Intraoral occlusal view of the implant immediately before prosthesis delivery at tooth number 46. (**B**): Intraoral lateral view of the implant immediately before prosthesis delivery. (**C**): Intraoral lateral view at one year after loading follow-up. (**D**): Periapical radiograph at prosthesis delivery. (**E**): Periapical radiograph at one year after loading follow-up.

**Figure 4 dentistry-09-00035-f004:**
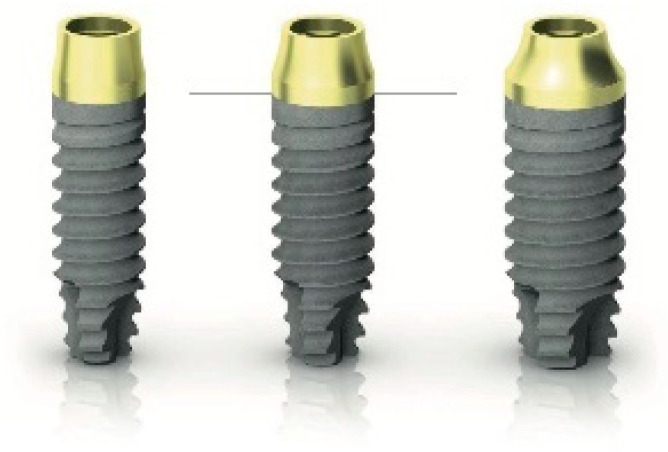
Reference point for marginal bone loss (MBL).

## Data Availability

Clinical data belonged to the authors and they are available upon request.

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
