# Peer review of "New Generation of Fixture–Abutment Connection Combining Soft Tissue Design and Vertical Screw-Retained Restoration: 1-Year Clinical, Aesthetics and Radiographic Preliminary Evaluation"

_dentistry, 2021, doi:10.3390/dj9040035_

Round 1
Reviewer 1 Report
Dear Authors,
It is a pleasure to become a reviewer of such well-designed research.
The topic is up to date and the structure of publication seems to be very clear and elegant as it fulfills the STROBE guidelines.
It was a great experience for me that you combine the restoration of tissue level implants with Biologically Oriented Preparation Technique, in this case rather Implant BOPT.
I have some inspirations or questions that do not diminish the importance of your publication.
- Do you try to build the statistically significant patient group or just try to present the preliminary study. If it was the first option it would be elegant to provide the information, how you measured the size of the group.
- You have described two cases with bleeding on probing whilst there were four patients with plaque. Were they the same patients? How do you try to explain that almost one third of your patients have unsatisfactory hygiene and does it affect the bleeding on probing?
- Do you consider the gingiva biotype in your patients?
- You noted that the maximum insertion torque in the time of implantation was 35 Ncm. Were there any variables in the torque (of course not exceeding 35 Ncm) as it is known as a important factor of future osseointegration, but if too high sometimes may provoke the marginal bone resorption?
- As for PES measurements, it seems to be a little tricky if aesthetic and non- aesthetic zone, upper and lower jaw implantation are performed and then assessed in one group. Do you recognize these variables?
Best regards
Author Response
Dear reviewer 1. Thanks for your comments. I improved the manuscript according to all of your suggestions. Following step-by-step responses to your comments.
- Do you try to build the statistically significant patient group or just try to present the preliminary study. If it was the first option it would be elegant to provide the information, how you measured the size of the group.
- No sample size was calculated. This information has been added. Preliminary evaluation has been clarified in the title and thought the text.
- You have described two cases with bleeding on probing whilst there were four patients with plaque. Were they the same patients? How do you try to explain that almost one third of your patients have unsatisfactory hygiene and does it affect the bleeding on probing?
- These points have been clarified. "One year after loading two out of 13 implants presented bleeding on probing (0.15 ± 0.38; corresponding to the 15.4% of the sites). These and other two (a total of four out of 13 patients) presented with plaque at the one year of follow-up (0.31 ± 0.48; corresponding to the 30.8% of the sites). A possible reason why almost one third of your patients have unsatisfactory hygiene could be that patients received delayed professional maintenance therapy due to COVID-19 pandemic."
- Do you consider the gingiva biotype in your patients?
- No. Thank you for your suggestion. This point has been added as limitation.
- You noted that the maximum insertion torque in the time of implantation was 35 Ncm. Were there any variables in the torque (of course not exceeding 35 Ncm) as it is known as a important factor of future osseointegration, but if too high sometimes may provoke the marginal bone resorption?
- We completely agree. Great attention has been placed to avoid excessive torque. This point has been explained: "Bone quality was intraoperative assessed and reported as hard, medium and soft. Drilling portal was adapted to the bone quality. The surgical unit was set with a torque of 35 Ncm during implant insertion. Back and forth movements was performed to place implants without exceed 35 Ncm."
- As for PES measurements, it seems to be a little tricky if aesthetic and non- aesthetic zone, upper and lower jaw implantation are performed and then assessed in one group. Do you recognize these variables?
- No, unfortunately I do not recognized variables due to the small sample size (already reported as limitation).

Reviewer 2 Report
This research is under the scope of this journal; the topic is relevant for readers, and this research deals with potentially significant knowledge to the field.
However, there are some concerns about the present manuscript:
Abstract.
- How many implants were placed? And what was the total of patients?
- In the results, is important to show more information, add some of the p-values.
Introduction
- What is the importance of this study for dental clinical? What is the gap in this field of literature?
- Which results are comparable with others study?
- What was the null hypothesis for this study?
Material and Methods
- How was the sample calculated? Did the authors perform a power analysis to evaluate if this sample size was appropriate?
- This section would better communicate with readers if restructured. A flowchart or diagram of the experimental processing would be valuable.
- Figures 1,2 and on page 7 line 225-226 - a group of the figures for the implants.
- In the case, Add the radiograph pre-implant, and identified the location of the implant in the legend.
Discussion
- Please, clarified what was the limitation of this study?
And also, clarified the future perspectives also add in the discussion.
Perhaps in a second study, authors must increase the number of participants and the time to collect data.
References
- Please insert in the text (all manuscript) before the end-point.
- References are not standardized.
- The titles of references have a different format, the title of the article is written in capital letters at the beginning of words, others only in lower case. Also, the standardized format of presentation in the journal's name. Because names have written in a different format, one is not abbreviated, others are not.
Author Response
Dear reviewer 2. Thanks a lot for your suggestions. I improved the manuscript as possible. Please check my responses.
REVIEW 2
This research is under the scope of this journal; the topic is relevant for readers, and this research deals with potentially significant knowledge to the field.
However, there are some concerns about the present manuscript:
Abstract.
- How many implants were placed? And what was the total of patients?
- Thank you. Patients/implats have been added "Overall, 13 patients (4 women and 9 man; mean age 50 ± 22 years) with same number of implants were treated and followed for one year after loading."
- In the results, is important to show more information, add some of the p-values.
- Due to this is a single cohort study, P values are not available. This has been reported as limitation. Now we has added all the information regarding this case.
Introduction
- What is the importance of this study for dental clinical? What is the gap in this field of literature?
- This point has been already reported, both in the introduction, as well as in the discussion section.
"In fact, the biomechanical aspects refer to masticatory load, as well as, pathological jaw movements (i.e. bruxism) as a major cause, while, the biological hypothesis considers the bacteria present at the level of the microgap between fixtures and abutments as main etiological agents. Hence, the need of introduce new prosthetic concepts is able to improve esthetics, and to reduce peri-implant bone loss. "
"The purpose of the current case series study was to examine a new type of screw-retained prosthesis delivered on tissue level implants with conical external vertical seal and internal hexagon connection."
- Which results are comparable with others study?
- According to the STROBE guidelines, this point has been reported in the discussion section.
- What was the null hypothesis for this study?
-Thank you for your comment. Due to this is a single cohort study, null hypothesis can not be considered
Material and Methods
- How was the sample calculated? Did the authors perform a power analysis to evaluate if this sample size was appropriate?
- Sample size calculation was not performed. According to the reviewer 1, this study as been changed as preliminary report.
- This section would better communicate with readers if restructured. A flowchart or diagram of the experimental processing would be valuable.
- Thank you! This section has been already structured according to the STROBE guidelines. A flowchart could be reported in the results section. nevertheless, due to the nature of the study (single cohort), a flowchart is not helpful for readers.
- Figures 1,2 and on page 7 line 225-226 - a group of the figures for the implants.
- Figures have been checked and organized.
- In the case, Add the radiograph pre-implant, and identified the location of the implant in the legend.
- Tooth position has been added.
Discussion
- Please, clarified what was the limitation of this study? And also, clarified the future perspectives also add in the discussion.
- Limitations of the study and further prospective has been improved, also according to the review 1 "Due to it was designed as a prospective case series study, the main limitations were the lack of a control group, and the small sample size. In addition, gingiva biotype was not considered in the present study. Hence, this research should be considered as a proof-of-concept study to act as a pilot for future multicenter randomized controlled trial with larger sample and longer follow-up."
Perhaps in a second study, authors must increase the number of participants and the time to collect data.
- Of course, we will take your suggestions and comments into further study. Thank you!
References
- Please insert in the text (all manuscript) before the end-point.
- References are not standardized.
- The titles of references have a different format, the title of the article is written in capital letters at the beginning of words, others only in lower case. Also, the standardized format of presentation in the journal's name. Because names have written in a different format, one is not abbreviated, others are not.
-Thanks. I checked all the reference. Titles was written according PubMed, References structured according to the Dentistry Journal. I hope they are fine. I defer to the editorial office for further corrections.

Round 2
Reviewer 2 Report
This research is under the scope of this journal; the topic is interesting for readers.
The authors improved the quality of the manuscript after the reviewer's indications. But, sill need improve the formatting of the figures with the text. The figures should be grouped into a single figure.
Author Response
Dear reviewer, thanks to approved our corrections in the first round.
Figures have been modified as suggested. Figure 1 has been changed. Figure 3 to 7 have been merged in only one. The text/figures has been checked accordingly.
Best regards, Irene.
This manuscript is a resubmission of an earlier submission. The following is a list of the peer review reports and author responses from that submission.